# Plasmon Driven Nanocrystal Transformation by Aluminum Nano-Islands with an Alumina Layer

**DOI:** 10.3390/nano13050907

**Published:** 2023-02-28

**Authors:** Xilin Zhou, Huan Chen, Baobao Zhang, Chengyun Zhang, Min Zhang, Lei Xi, Jinyu Li, Zhengkun Fu, Hairong Zheng

**Affiliations:** 1School of Physics and Information Technology, Shaanxi Normal University, Xi’an 710119, China; 2School of Electronic Engineering, Xi’an University of Posts & Telecommunications, Xi’an 710121, China

**Keywords:** surface plasmon, plasmonic photothermal, rare earth doped nanocrystal, wide-range absorption

## Abstract

The plasmonic photothermal effects of metal nanostructures have recently become a new priority of studies in the field of nano-optics. Controllable plasmonic nanostructures with a wide range of responses are crucial for effective photothermal effects and their applications. In this work, self-assembled aluminum nano-islands (Al NIs) with a thin alumina layer are designed as a plasmonic photothermal structure to achieve nanocrystal transformation via multi-wavelength excitation. The plasmonic photothermal effects can be controlled by the thickness of the Al_2_O_3_ and the intensity and wavelength of the laser illumination. In addition, Al NIs with an alumina layer have good photothermal conversion efficiency even in low temperature environments, and the efficiency will not decline significantly after storage in air for 3 months. Such an inexpensive Al/Al_2_O_3_ structure with a multi-wavelength response provides an efficient platform for rapid nanocrystal transformation and a potential application for the wide-band absorption of solar energy.

## 1. Introduction

In recent years, there has been increasing interest in the plasmonic photothermal effects of metal nanostructures, which are widely used in a variety of applications including biomedical therapy [1,2,3,4], photocatalysis [5,6,7,8], photothermal imaging [9,10] and optofluidic technology [11,12]. Oscillations of charge carriers in plasmonic metal nanoparticles activated by resonant absorption of light are accompanied by local temperature increase due to nonradiative plasmon damping, and the heat generated is subsequently transferred to the surrounding medium; the whole process typically occurs at timescale of nanoseconds [13,14,15]. According to the surface plasmons decaying process, the plasmonic photothermal effects have the characteristics of efficient photothermal energy conversion, fast heat production rate, and the generation of extremely high temperatures on the nanoscale. It is important to effectively use the thermal energy from surface plasmons light absorption. In addition, plasmonic photothermal effects are strongly dependent on the intrinsic frequency of the metal nanostructures and the excitation wavelength. Metal nanostructures with narrow plasmon linewidths can only respond to a specific wavelength, forming a photothermal mode and reducing the efficiency of light utilization, which limits their applications such as solar energy harnessing [16,17].

The basic strategy to change the spectral absorption and improve the efficiency of photothermal conversion is to select suitable metal materials and modify the construction of nanostructures [18,19]. Aluminum nanomaterials possess a much wider optical range of localized surface plasmon resonances (LSPR) than gold, silver and copper, from the ultraviolet to the near-infrared region. In addition, its low cost, high natural abundance, and ease of processing make Al a sustainable plasmonic material [20,21]. Since aluminum is susceptible to oxidation and the resulting degradation of optical properties, a thin oxide layer formed rapidly on the surface of high-purity aluminum is important to protect it from further oxidation and contamination, and to improve durability. Recently, it has been reported that photothermal efficiency can be improved by fabricating aluminum micro/nano-structures to achieve anti-icing/deicing and solar water desalination [22,23]. However, the preparation of these broadband absorbing structures demands a complex nanofabrication process and often relies on sophisticated equipment, further limiting their large-scale production. It is valuable for practical applications to fabricate aluminum nanostructures with high plasmonic property by simple, rapid and low-cost technology processes.

In this work, simply prepared self-assembled Al NIs with an alumina layer are selected as a plasmonic photothermal substrate to achieve a fast transformation of rare-earth-doped nanocrystals. It has been found that Al NIs with an alumina layer can respond to several different wavelength lasers simultaneously, and they have a wide range of absorption, from visible to near-infrared light. With the increase in the thickness of the alumina heat-trapping layer, the heat utilization is further enhanced, thus significantly improving the transformation efficiency of the nanocrystals. Additionally, the Al/Al_2_O_3_ structure still produces efficient nanocrystal transformation even in low temperature environments. Additionally, the alumina layer on the Al NIs surface prevents further oxidation of the aluminum and keeps the crystal transformation efficiency stable in air for at least three months.

## 2. Materials and Methods

The substrates used are glass, which are cleaned before loading into the high vacuum coating. It consists of soaking glass substrates with piranha solution for 24 h and then ultrasonically cleaning with alcohol, acetone and deionized water for half an hour. Piranha solution is a mixture of 98% concentrated sulfuric acid and 30% hydrogen peroxide solution according to the volume ratio of 3:1. Due to its own strong oxidizing property, the solution can be used to remove organic residues from the glass substrate. NaYF_4_:Eu^3+^ was prepared by a wet-chemical method. The raw materials were NaF (98%), Y(NO_3_)_3_ (99.9%), and Eu(NO_3_)_3_ (99.9%). All reagents were purchased from Sigma–Aldrich Chemicals Co. (Shanghai, China) First, suitable stoichiometric proportions of NaF, Y(NO_3_)_3_ and Eu(NO_3_)_3_ were dissolved in a conical flask with an appropriate amount of deionized water to form a mixture, and then the complex solution was heated at 75 °C for 2 h. After the temperature was cooled down, the solution was centrifuged and washed with deionized water and ethanol twice, forming the white precipitated product. The white precipitated product was then dried at 55 °C for 12 h to obtain the target material.

To characterize the sample morphologies, scanning electron microscopy (SEM) images were obtained with a FEI-Nova NanoSEM 450 at 10 kV, and atomic force microscopy (AFM) images were obtained with a Bruker-JPK Nano Wizard Ultras (Karlsruhe, Germany). The X-ray diffraction (XRD) pattern was obtained by using a Bruker D8 Advance diffractometer (Karlsruhe, Germany). UV–Vis spectra were acquired with a PerkinElmer Lambda 950 spectrometer (Waltham, MA, USA). Conventional bright-field TEM images were obtained with a Thermo Fisher Talos F200i (Waltham, MA, USA) operated at 200 kV. In situ laser irradiation and luminescence spectra measurements were conducted with a Lab Ram HR Evolution Raman system with a 100× (NA = 0.9) objective. To avoid any crystal transformation during luminescence spectral acquisition, a low laser power (0.2 mW) was used to obtain luminescence emission.

## 3. Results and Discussion

Figure 1a illustrates the configuration of a wide-range laser wavelengths response photothermal system, which is excited by a multi-wavelength laser to produce heat for crystal transformation. Using Al/Al_2_O_3_ with a broadband plasmon resonance activity as a plasmonic heat source, the excitation of a rare-earth-doped nanocrystal placed on top with a multi-wavelength laser can lead to rapid nanocrystal transformation. Figure 1b shows the preparation process of the plasmonic photothermal system. Firstly, aluminum is deposited onto preprocessed glass substrates by thermal evaporation under a high vacuum, and then annealed at 300 °C under argon atmosphere to form aluminum nanoparticle arrays (Al NIs films). Then, Al_2_O_3_ layers with different thicknesses are deposited onto the Al NIs by atomic layer deposition (ALD). Finally, rare-earth-doped luminescent nanocrystals (NaYF_4_:Eu^3+^) are deposited on the Al/Al_2_O_3_ structure by dropwise addition.

Compared with lithography and chemical synthesis, metal evaporation followed by short-term thermal annealing is a simple, fast and low-cost method and allows the formation of well-separated nanostructures over large areas [24]. Figure 2a is the SEM image of aluminum deposited on Si substrate with annealing at 300 °C, where Al NPs are well separated to form an array of aluminum nanoparticles. As shown in the AFM image in Figure 2b, the root mean square value of the surface roughness of Al NIs is 3.741 nm. Due to the AFM tip-sample convolution effect, the nanostructured particles in the AFM image are larger in size compared to the SEM. During AFM scanning, the tip-sample convolution effect is caused by the geometrical interactions between the tip and surface features being imaged. The tip-sample convolution effect is one of the main causes of AFM artifacts, owing to the finite sharpness and characteristic geometry of tips [25]. In addition, aluminum films of the same deposited thickness are annealed at different temperatures, and as seen from the optical and AFM images the average size and gap of the particles do not change significantly with increasing annealing temperatures (Appendix A). It is the spontaneous oxidation of aluminum that produces the alumina layer, which improves thermal stability, effectively increasing the heat resistance of the aluminum film and limiting the migration rate of aluminum atoms during the annealing treatment [26,27]. In the subsequent experiments, Al NIs films annealed at 300 °C are selected as the substrates. Next, the Al_2_O_3_ layer is uniformly deposited on the surface of the Al NIs. The SEM image of the cross-section of the Al/Al_2_O_3_ structure shows that the thickness of Al_2_O_3_ is approximately 52 nm (Figure 2c). More detailed information of the Al/Al_2_O_3_ structure is shown in Appendix A.

Rare-earth-doped luminescent nanocrystals (NaYF_4_:Eu^3+^) are prepared by a wet-chemical method (see Materials and Methods). The SEM image in Figure 3a shows that the NaYF_4_:Eu^3+^ particles are in the shape of flowers, with uniform size and good dispersion. A single nanoflower particle size of about 500 nm is shown in the TEM image in Figure 3b, and the suitable particle size and good dispersion are favorable for the subsequent photothermal study of the surface plasmon induced nanocrystal transformation of the metal nano-islands films. The high-resolution TEM image in Figure 3c shows that the NaYF_4_:Eu^3+^ nanoflower consists of many small grains, and the crystallinity of the sample is poor because the sample was not treated at high temperature during the synthesis process. In addition, the XRD pattern shows that NaYF_4_ is a cubic phase, and the intensity and sharpness of the diffraction spectrum of the sample are slightly lower, indicating the poor crystallinity of the sample (Appendix A). The EDX elemental mapping of the nanoflower in Figure 3d shows the presence of Na, Y, F and Eu. 

The luminescence spectrum of doped Eu^3+^ is used to monitor the crystal transformation of the matrix materials [28,29]. A low power (0.2 mW) 532 nm laser is used as the excitation source to obtain the luminescence spectrum, and a high power (23 mW) 976 nm laser is used as the irradiation light to drive the crystal transformation. The in situ luminescence spectra of a single nanoflower before and after 976 nm laser irradiation are shown in Figure 4a. Wide bands of luminescence centered at 590 nm, 615 nm and 700 nm with weak intensity are observed, which indicates the poor crystallinity of the initial NaYF_4_. After laser irradiation, there is a sharp band at 610 nm with strong luminescence intensity, indicating that the nanocrystal has better crystallinity. From the inset in Figure 4a, it can be seen that the single flower-like NaYF_4_ nanocrystal is transformed in situ into a spherical particle. The distribution of Y, O and Al elements in the product shows that the NaYF_4_:Eu^3+^ particle transformed to Y_2_O_3_:Eu^3+^ after laser irradiation (Figure 4b). More details on the product can be found in previous work [8]. Because of the short relaxation time and the high temperature of the plasmonic photothermal effects, it is difficult to detect and measure the heat production by conventional methods. On the other hand, thermal effects play an important role in the plasmon driven crystal transformation, and hot electrons play an auxiliary and synergistic role. Therefore, the irradiation time of the plasmon-driven rare-earth-doped luminescent nanocrystal transformation is used to estimate the plasmonic photothermal effects. The rate of the above crystal transformation can be easily controlled by the power of laser irradiation. As shown in Figure 4c, the irradiation time required for the transformation to Y_2_O_3_ depends on the laser power. As the laser power increases from 10 mW to 22 mW, the irradiation time decreases from approximately 1.5 s to 40 ms for 976 nm with the Al/Al_2_O_3_ structure, indicating that the plasmonic photothermal effects improve. With the increase in laser intensity, the electromagnetic field of surface plasmons can be enhanced in the same proportion, and the density and thermal effect of hot electrons caused by electromagnetic field attenuation can also increase, thus improving the transformation efficiency of the luminous crystals.

As shown in Figure 5a, the response of the Al/Al_2_O_3_ structure to different laser wavelengths of 532 nm, 633 nm and 976 nm is investigated, and all these laser irradiations can induce structural transformations in the crystals. In Figure 5b, by using Al NIs without an Al_2_O_3_ layer as the photothermal substrate, it is found that 532 nm, 633 nm and 976 nm laser irradiation could induce crystal structural transformation, but 532 nm and 633 nm as the irradiation laser could not induce the complete crystal transform to Y_2_O_3_. Figure 5c shows that both Al NIs and the Al/Al_2_O_3_ structure have absorption in the range of 350–1100 nm, and the intensity of the Al/Al_2_O_3_ structure is even weaker. These three different wavelengths of laser irradiation can all induce collective resonant behavior of the electrons and generate heat to drive the transformation of the rare earth nanocrystals. The introduction of alumina can effectively improve the plasmon-induced photothermal conversion efficiency.

The difference occurs because the Al_2_O_3_ layer alters the polarizability and absorption cross-section of the aluminum nanoparticles, increasing the heat production efficiency [30,31]. On the other hand, the alumina layer has a relatively higher thermal conductivity than the glass substrate. Therefore, the heat transfer is more favorable through the alumina layer rather than through the air and the glass substrate. Figure 6a shows a heat transfer schematic diagram with and without the Al_2_O_3_ heat trapping layer. The arrow points in the direction of the heat flow. A large amount of heat generated by the aluminum nanoparticles around the sample can be transferred to it via the Al_2_O_3_ layer, reducing heat diffusion into the air and the glass substrate, leading to an increase in the temperature of the nanocrystals and an enhanced crystal transformation efficiency. In order to further investigate the dependence of the Al_2_O_3_ layer enhanced crystal transformation efficiency, the crystal irradiation time as a function of Al_2_O_3_ thickness is shown in Figure 6b. The irradiation laser used in the experiment is 976 nm (23 mW). Firstly, the required irradiation time is about 600 ms at a thickness of 5 nm Al_2_O_3_ and then decreases gradually within the thickness range from 5 to 40 nm and finally plateaus at 40 ms from 40 to 70 nm. Initially, as the thickness of the alumina increases, it enhances the heat transfer to the crystal and raises the crystal temperature rapidly; later, as the thickness further increases, the mild change in crystal temperature is not sufficient to affect the crystal transformation time. In addition, NaYF_4_:Eu^3+^ was placed on a smooth glass or on only a 50 nm Al_2_O_3_ layer, and the particles were irradiated with a 976 nm laser for 30 min (Appendix A). The nanoflower spectra hardly changed before and after irradiation, indicating that the heat driving the nanocrystal transformation mainly comes from the metal under laser irradiation.

With the Al_2_O_3_ heat-trapping layer, plasmon driven nanocrystal transformation is also realized in low temperature environments. As shown in Figure 7a, the required laser irradiation time increases from 2 s to 36 s with a decreasing temperature from 20 to −60 °C. Although a longer irradiation time is required in low temperature environments, nanocrystal transformation still occurs in an acceptably short time. In addition, both Al NIs and the Al/Al_2_O_3_ structure can be stored stably in air for at least three months. As shown in Figure 7b, the required transformation times for Al NIs with and without Al_2_O_3_ are both well stabilized. Al oxides rapidly in air and forms a thin self-limiting oxide layer. Because of the compact texture of alumina, it protects the metal inside from further oxidation, making the plasmonic photothermal effects stable. Therefore, the plasmonic photothermal system has very good heat production in low temperature environments and long-term storage stability in air, which make it more suitable for practical applications.

## 4. Conclusions

In summary, a fast crystal transformation from polycrystalline NaYF_4_ nanoflowers to globular crystal Y_2_O_3_ is realized, with the self-assembled Al NIs with a thin Al_2_O_3_ layer as a plasmonic photothermal substrate. It has been demonstrated that the Al/Al_2_O_3_ structure can be excited by lasers with a wide range of wavelengths. The Al_2_O_3_ heat-trapping layer is shown to facilitate the coupling of heat generated near the Al nano-islands into the NaYF_4_:Eu^3+^ nanoflowers. The increased thickness of the Al_2_O_3_ layer enhances the efficiency of the heat transfer, resulting in a faster crystal transformation rate. In addition, the Al/Al_2_O_3_ structures show excellent photothermal effects even in low-temperature environments and can be preserved in air for at least three months. The low cost, simple preparation process, broad plasmon resonance activity and excellent stability make the Al/Al_2_O_3_ structure potentially useful in solar thermal conversion.

## Figures and Tables

**Figure 1 nanomaterials-13-00907-f001:**
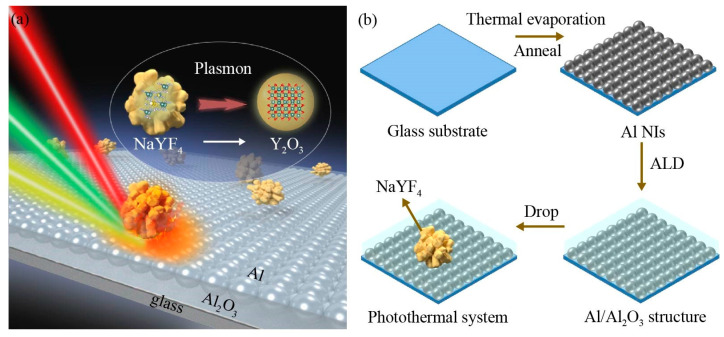
(**a**) Schematic illustration of the fast transformation of nanocrystals driven by the plasmonic photothermal system with a multi-wavelength response. (**b**) Preparation process of the plasmonic photothermal system.

**Figure 2 nanomaterials-13-00907-f002:**
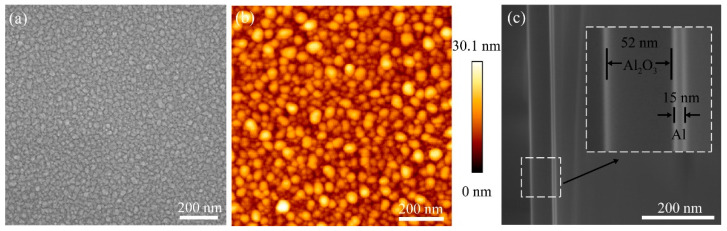
(**a**) SEM image of the Al NIs with annealing at 300 °C. (**b**) AFM image of the Al NIs annealed at 300 °C on a glass substrate. (**c**) SEM image of a cross-section of the Al/Al_2_O_3_ structure.

**Figure 3 nanomaterials-13-00907-f003:**
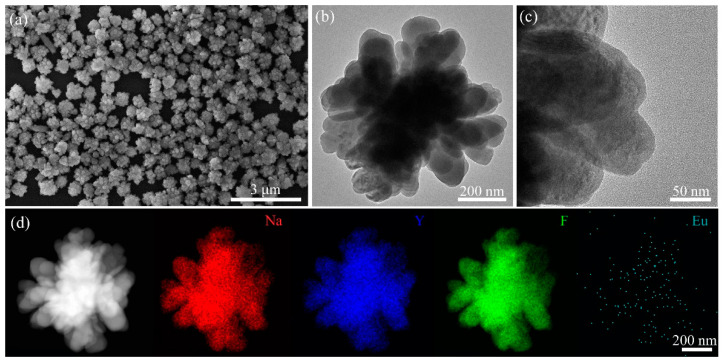
(**a**) SEM image, (**b**) TEM image, (**c**) enlarged high-resolution TEM image, (**d**) HAADF-STEM image and EDX elemental mapping of the NaYF_4_:Eu^3+^ flower-shaped nanocrystals.

**Figure 4 nanomaterials-13-00907-f004:**
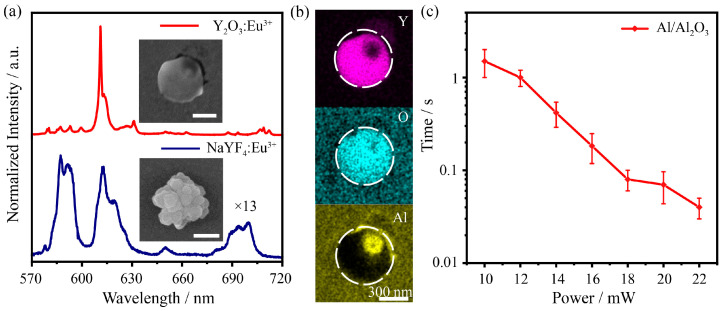
(**a**) In situ luminescence spectra of a Eu^3+^-doped single nanoflower before and after 976 nm laser irradiation, and inserted SEM images show the initial and transformed nanocrystal. Scale bar, 300 nm. (**b**) EDX elemental mapping of the spherical product. (**c**) Laser power dependent irradiation time.

**Figure 5 nanomaterials-13-00907-f005:**
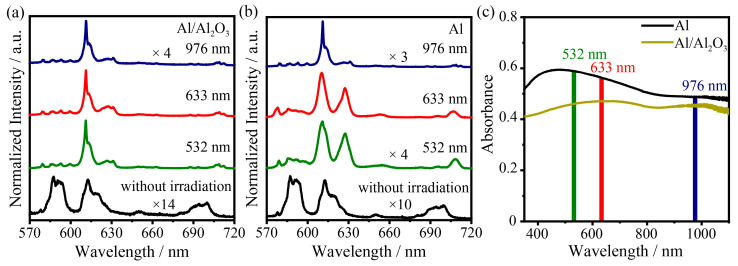
Luminescence spectra of the (**a**) Al/Al_2_O_3_ structure and (**b**) Al NIs plasmon driven nanoflower transformation after laser irradiation at 532, 633, and 976 nm. (**c**) UV–Vis spectra of Al NIs and Al/Al_2_O_3_ structure.

**Figure 6 nanomaterials-13-00907-f006:**
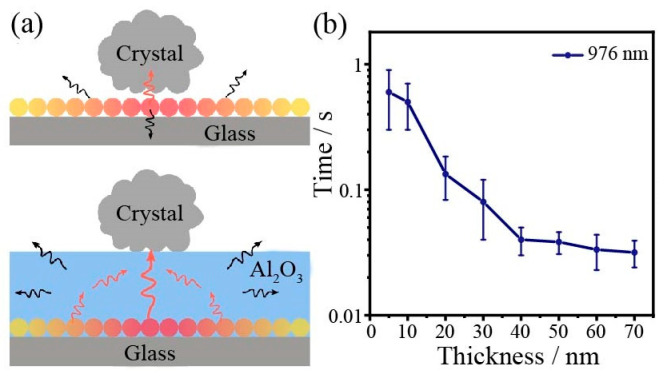
(**a**) Schematic diagram of heat transfer in the system with and without the Al_2_O_3_ heat trapping layer. (**b**) Crystal irradiation time for Al NIs as a function of Al_2_O_3_ thickness, from 5 to 70 nm.

**Figure 7 nanomaterials-13-00907-f007:**
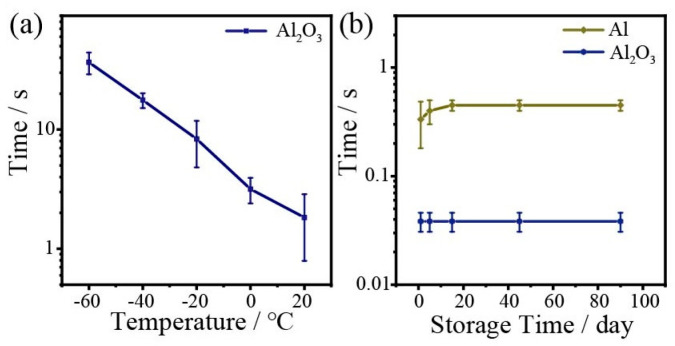
(**a**) Irradiation time in low-temperature environments with a laser power of 21.5 mW at 976 nm. (**b**) The stability of irradiation time for Al NIs and the Al/Al_2_O_3_ (50 nm) structure.

## Data Availability

Data are contained within the article or Appendix A.

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
