# Peer review of "Plasmon Driven Nanocrystal Transformation by Aluminum Nano-Islands with an Alumina Layer"

_nanomaterials, 2023, doi:10.3390/nano13050907_

Round 1

Reviewer 1 Report

Dear Authors,

I have read your manuscript carefully and believe that it needs minor corrections. I have not noticed any ambiguities as far as the scientific side of the manuscript is concerned. I only have doubts about Figure 2 where parts A and B show SEM and AFM respectively. Why is the grain in both images so different? The nanostructures from the AFM measurement seem to be several times larger.

Further remarks concern:

1. The Conclusions chapter does not contain conclusions from the research, but only a summary of the research carried out.

2. The Author Contributions section does not contain a description of the distribution of the authors' contributions, but only a declaration of reading the manuscript by the authors and their consent to publication.

3. English should be improved. Some sentences are difficult to understand and others are completely grammatically incorrect. for example, the sentence starting on line 209 ("When the irradiated laser is 976 nm ....", and the sentence starting on line 165 ("On the other hand ....".

Author Response

Please see details in the attached file.

Reviewer 2 Report

The paper accounts for utilizing self-assembled Al nanoparticles covered with an Al/Al2O3 layer as a plasmonic photothermal substrate. It was shown that this system can be excited by a wide range of laser wavelengths and act as a plasmonic photothermal structure. The controllability of plasmonic photothermal nanocrystal transformation was demonstrated with NaYF4:Eu3+ nanoflowers transformed to single crystal Y2O3 dots by using the characteristic luminescence spectra of the two types of NPs. The Al2O3 heat-trapping layer proved advantageous for coupling the heat generated in the vicinity of the Al nanoislands into the NaYF4:Eu3+ nanoflowers. Moreover, the speed of the heat transfer could be controlled by the thickness of the alumina layer, and the process worked in a low-temperature environment as well. The results suggest that the Al NP – Al2O3 structure can have a potential in solar thermal conversion.

Plasmonic NPs and rare-earth-doped NPs have vast literature because of their enormous versatility of applications. The authors themselves have published a previous paper which discussed the plasmonic heat transformation of the NaYF4:Eu3+ nanoflowers in the presence of Au NPs [8]. The present paper’s strengths and novelty are that Al NP – Al2O3 structure is technologically simple to produce and that the protective alumina layer plays an essential role in the control of heat transfer.

My questions and comments are the following:

1.      Could the authors compare the efficiency of heat transfer to the NaYF4:Eu3+ nanoflowers in the system described in [8] (i.e. Au NPs scattered over the nanoflowers’ surface) and in the present Al NP – Al2O3 –nanoflowers system?

2.      Could you characterize the AL NI size distribution and homogeneity? Is the apparent difference in the sizes of the Al Nis in Figure 2. (a) and (b) due to the AFM tip-sample convolution effect?

3.      The 15 nm thick Al layer seems to be completely homogeneous in Figure 2. (c). Why can’t the different Al NIs be seen at least along the vertical direction? Based on (a) and (b), one would expect several NIs in the shown range.

4.   Line 168, Figure 4 (c): Please explicitly define what you mean by “the irradiation time required for the transformation”.

5.      Figure 4 (a), Figure 5 (a), (b): The vertical axis of the graphs uses arbitrary units. The intensities of the spectra with and without irradiation seem to be similar. Is it the case?

6.      Figure 5 (c): Using different colour curves for Al and Al/Al2O3 would make the figure more self-explanatory.

In summary, the submitted manuscript is scientifically sound and presented clearly. It is of interest to the scientific community. I recommend its publication after minor revision. 

Author Response

Please see the details in the attached file.

Round 2

Reviewer 1 Report

Dear Authors,

Thank you for the answers and kind explanations. In my opinion the manuscript does not need further corrections.

Author Response

We thank the reviewer for the positive assessment and valuable suggestions.